# Dreaming in Parasomnias: REM Sleep Behavior Disorder as a Model

**DOI:** 10.3390/jcm11216379

**Published:** 2022-10-28

**Authors:** Elisabetta Fasiello, Serena Scarpelli, Maurizio Gorgoni, Valentina Alfonsi, Luigi De Gennaro

**Affiliations:** 1Department of Psychology, Sapienza University of Rome, Via dei Marsi 78, 00185 Rome, Italy; 2Body and Action Lab IRCCS Fondazione Santa Lucia Foundation, 00179 Rome, Italy

**Keywords:** parasomnias, REM sleep behavior disorder, dream recall frequency, dream contents, oneiric activity

## Abstract

Sleep parasomnias have drawn the interest of sleep experts because they represent a valuable window to directly monitor dream activity and sleep mentation associated with nocturnal events. Indeed, parasomnias and their manifestations are helpful in investigating dream activity and features, overcoming methodological limits that affect dream study. Specifically, REM sleep Behavior Disorder (RBD) is a parasomnia characterized by enacted dream episodes during Rapid Eye Movements (REM) sleep, caused by the loss of physiological atonia. Patients suffering from RBD report a peculiar oneiric activity associated with motor episodes characterized by high Dream Recall Frequency (DRF) and vivid dreams. Additionally, isolated RBD (iRBD) represents a prodromal stage of neurodegeneration preceding the development of α-synucleinopathies. This narrative review aims to combine evidence describing dream activity in RBD and similarities and differences with other NREM parasomnias. Moreover, a special focus has been reserved for those conditions in which RBD is associated with α-synucleinopathies to clarify the potential role of dreams in neurodegenerative processes.

## 1. Introduction

Sleep parasomnias are described in the third edition of the International Classification of Sleep Disorders (ICSD-3) [1] as sleep disorders involving unusual motor and vocal behaviors accompanied by emotional or sensory perceptions and associated with dream mentation. These episodes appear during transition periods between sleep and wake or are concomitant to specific sleep stages. Hence, parasomnias can be classified into Non-Rapid Eye Movement (NREM) (i.e., confusional arousals, Sleep Walking (SW), Sleep Terrors (ST), and sleep-related eating disorder) and Rapid-Eye Movement (REM) related (i.e., REM sleep Behavior Disorder (RBD), recurrent isolated sleep paralysis, nightmare disorder, and sleep-related hallucinations) [1].

REM and NREM parasomnias have drawn the interest of sleep experts not only on the clinical characteristics reported by the patients but also because these conditions represent a precious window to directly monitor dream activity and sleep mentation associated with nocturnal events. Indeed, an intrinsic issue of dream study concerns their inaccessible nature: dream contents are not directly accessible, and knowledge on oneiric activity is collected through retrospective recall [2]. As a consequence, the retrospective nature of dream collection leads to several methodological problems due to distortions and omissions in the recall caused by memory reprocessing [3].

In this review, we focus on RBD as privileged parasomnia to explore the dream process. Indeed, this is a sleep disorder characterized by enacted dream episodes during REM sleep, caused by the loss of physiological atonia. Clinical observations have pointed out in RBD a peculiar oneiric activity associated with motor episodes characterized by high Dream Recall Frequency (DRF) and vivid dreams, containing elements of violence and attacks (by people and animals) [4]. Moreover, RBD may present itself as isolated (iRBD), namely the parasomnia is not due to other neurological conditions, and it represents a prodromal stage of neurodegeneration. Specifically, longitudinal studies showed in iRBD patients an elevated risk of developing α-synucleinopathies (i.e., Parkinson’s Disease (PD), Dementia with Lewy Bodies (DLB), and Multiple System Atrophy (MSA)) that seems to increase over several years (i.e., 33.5% at five years after diagnosis, 82.4% at 10.5 years, and 96.6% at 14 years) [5]. In light of this strong association between iRBD and neurodegenerative diseases, in the last years, scientists have researched neuropsychological, electrophysiological, and neuroimaging biomarkers for a timely prediction of phenoconversion [6]. In this view, oneiric activity in RBD has been proposed as associated with biological processes leading to α-synucleinopathies. Indeed, evidence in PD patients, already in the early stages of the disease, shows a high prevalence of distressing and vivid dreams, dreams with violent content, and nightmares [7], supporting the notion that RBD patients share similar dream features with PD patients [8].

Within this theoretical background, this paper aims to combine evidence describing dream activity and its features in RBD, and similarities and differences with other NREM parasomnias, such as SW and ST. Moreover, a special focus has been reserved for those conditions in which RBD is associated with α-synucleinopathies to clarify the role of dreams in neurodegenerative processes.

## 2. Materials and Methods

### 2.1. Search Strategy

We conducted a literature search from March 2022 to July 2022, following the Preferred Reporting Items for Systematic Reviews and Meta-Analyses (PRISMA) statement. The literature search was conducted using PubMed, Scopus, PsyArticles, and Web of Science, considering available studies up to July 2022. Search terms included: “Rem Sleep Behavior Disorder”, “dream enactment”, “dream”, “dream recall”, “dreaming”, “oneiric”, “nightmare”, and “dream report”. Search terms had to be contained in the title, abstract, and/or keywords.

### 2.2. Inclusion/Exclusion Criteria

Titles, abstracts, and keywords were inspected to meet the following inclusion criteria: (1) English language; (2) peer-reviewed article; (3) cross-sectional or longitudinal design; (4) main focus on at least one of the investigated phenomena (RBD sample, with or without comorbidities; dreams; nightmares); (5) quantitative/qualitative examination of at least one aspect of the investigated phenomena (frequency, qualitative features, content); (6) the method employed to diagnose RBD had to include at least one night of video polysomnographic (vPSG) registration. Books, abstracts, comments, reviews, meta-analyses, pre-prints, and letters to editors were excluded.

One expert researcher chose eligible articles through a multi-step process (title reading, abstract, and full-text assessment).

## 3. Results

Forty-one papers published between 1999 and 2022 met the inclusion criteria and were selected for our review. Thirty-two studies had a cross-sectional design, of which eight with a descriptive approach (i.e., between-group analyses were not performed), five papers adopted a longitudinal approach, two were retrospective studies, two adopted a multiple-awakenings protocol, and two papers selected were case series studies (see Table 1, Table 2, Table 3 and Table 4).

In all included studies, a total of 1936 RBD are enrolled. Moreover, among studies that considered RBD secondary to other pathologies, a total of 174 RBD patients were secondary to PD, 24 PD patients presented probable RBD, 15 RBD secondary to a Post-Traumatic Stress Disorder (PTSD), and 13 to DLB. The examined studies also considered a total of 614 Healthy Controls (HCs), 104 PD patients, 113 SW and ST sufferers, 64 with Obstructive Sleep Apnea (OSA), 13 DLB, and seven with PTSD. Moreover, a single study considered RBD and REM Sleep Without Atonia (RSWA) in children and adolescents [9].

Among RBD, idiopathic and secondary, 87.2% were men, reflecting the male predominance of this parasomnia [10]. Moreover, considering all papers, the mean age was 63.6 years, ranging from three years [9] to 88 years [11].

From a methodological point of view, the studies selected can be divided into two categories according to the dream assessment, as shown in Figure 1: retrospective (*n* = 32) and prospective (*n* = 11). Studies in both categories extracted quantitative (dreams and nightmares recall frequency) and qualitative (dream contents and themes) dream features through different procedures.

### 3.1. How RBD Patients Dream?

From the first observation and description [53], RBD drew the attention of sleep experts for its unique characteristics of dream-enactment during REM sleep.

Motor behaviors observed in this parasomnia appear to act out dream contents and settings, as demonstrated by the correspondence between the features of dream recall and the observed behaviors. Moreover, dream contents reported have recurrent elements and similar characteristics among patients. The analysis of dream contents (See Table 1) highlighted recurrent unpleasant dreams and nightmares reported by RBD patients [12,44]. Specifically, the main themes collected in the dream recall were attacks by people or animals [13,14,44], violence [11,12,15], and fright [12] (for detailed dream report examples, see Leclair-Visonneau et al. [45] in Table 1). Interestingly, these violent and aggressive oneiric themes are not due to and do not match personality features. Indeed, RBD patients did not show hostile and violent traits during wakefulness; on the contrary, they appeared quiet and calm [16,17,46]. Moreover, dream assessment proved that the more dreams with aggression, misfortune, and negative emotions occurred, the more the patients had lower traits of hostility, anger, and less tendency to be aggressive [16].

**Table 1 jcm-11-06379-t001:** Sample, design, tools, and findings in studies investigating dreaming in RBD and RBD vs. HC. Legend: X = absence of HC sample.

Study	RBD Sample	HC Sample	Design	Dream Measures	Main Findings
[11]	39 RBD < 50 y Mean age (SD): 32 ± 9 y Gender: 23 M/16 F 52 RBD ≥ 50 y Mean age (SD): 67 ± 8 y Gender: 39 M/13 F	X	Cross-Sectional	Not specified	**Dream Content:** Vivid dreams with violent content ∘ RBD < 50 y: *n* 39 RBD > 50 y: *n* 51 Sports dreams ∘ RBD > 50 y: *n* 1
[12]	20 early onset RBD Gender: 12 M/11 F 67 late-onset RBD Gender: 51 M/16 F	90 HC Gender: 63 M/27 F	Cross-Sectional	RBDQ-HK	**DRF:** In total, 91% RBD reported dreams more than 3 times per week **Dream-related scores (Factor1):** Absence of significant difference between RBD and HC Absence of significant difference between RBD with early and late-onset Vivid dreams Disturbances associated: in 54% RBD **Dream content in RBD:** Nightmares: 94% Violent or frightening dreams: 80%
[44]	4 iRBD Gender: 2 M/2 F	X	Cross-Sectional Descriptive	Not specified	**Dream content:** Unpleasant dreams such as being attacked (*n* 4) Attacked by someone (*n* 3) Arguing with someone (*n* 3) Chased by someone (*n* 2) Falling from a cliff (*n* 3) Attacked by an animal (*n* 2) ∘ Dog (*n* 1) ∘ Snake (*n* 1) Action-filled sports (*n* 1) ∘ Football (*n* 1) ∘ Ski (*n* 1) Children in life-threatening situation (*n* 0)
[13]	93 RBD Mean age: 64.4 y Gender: 81 M/12 F	X	Retrospective Descriptive Case Series	Not specified	**Dream Report** (*n* 67): Dreams associated with RBD activity (*n* 62; 93%) **Dream content described** (*n* 37; 55%): Defense against attack by people (57%) or animals (30%) Adventure dreams (9%) Sports dreams (2%) Aggression by the dreamers (2%)
[14]	203 iRBD Gender: 162 M/41 F	X	Longitudinal Descriptive	Semi-structured interview	**Unpleasant dream recall:** Present in 92.6% (nightmares) Absent in 7.4% Absence of significant differences between M and F RBD At the follow-up: <in the frequency and severity for RBD + OSAS treated with CPAP **Dream content:** Attacked by someone in 76.8% ∘ >in M RBD compared to F RBD (*p* < 0.001) Attacked by an animal in 39.9% Chased by someone in 55.7% Arguing with someone in 63.5% ∘ >in M RBD compared to F RBD (*p* = 0.003) Children in a life-threatening situation in 12.8% ∘ <in M RBD compared to F RBD (*p* < 0.001) Falling from a cliff in 47.8% ∘ <in M RBD compared to F RBD (*p* = 0.032) Action-filled sports in 15.8% ∘ >in M RBD compared to F RBD (*p* = 0.002)
[15]	7 RBD Gender: 5 M/2 F	X	Cross-Sectional Descriptive	Telephone interview	All RBD patients reported violent dreams
[45]	56 RBD Mean age (SD): 64.7 ± 8.2 y Gender: 43 M/13 F	17 HC Mean age (SD): 62.2 ± 7.1 y Gender: 14 M/3 F	Cross-Sectional Descriptive	Immediate dream recall through interview	**Detailed dream reports examples in the paper:**
[16]	49 RBD Mean age (SD): 67.5 ± 7.5 y Gender: 36 M/5 F	35 HC Mean age (SD): 69.1 ± 5.9 y Gender: 30 M/5 F	Cross-Sectional	Free recall and semi-structured interview scored by HVdC	**DRF:** >in RBD (*p* < 0.001) **Dream content** Aggression/Friendliness: >in RBD compared to HC (*p* < 0.001) Dreamer as aggressor: >in RBD compared to HC (*p* = 0.002; uncorrected) Dreams with at least one aggression: >in RBD compared to HC (*p* < 0.001) Animal: >in RBD (*p* = 0.00013) Familiar characters: <in RBD compared to HC (*p* = 0.065; uncorrected) Dreams with at least one sexual experience: <in RBD compared to HC (*p* < 0.001) Negative emotion: >in RBD compared to HC (*p* = 0.003; uncorrected) Male/female characters ratio: <in sRBD compared to iRBD (*p* < 0.00001) Striving: <in sRBD compared to iRBD (*p* < 0.001) **Correlations between dream and sleep/psychological features in RBD:** Positive correlation between A/C ratio and PLMI Positive correlation between % of dreams with aggression and PLMI Negative correlation between % of dreams with aggression and the Hostility AQ subscale Negative correlation between % of dreams with misfortune and the Anger AQ subscale Negative correlation between % of Negative emotion and the Physical Aggression AQ subscale
[18]	94 RBD Mean age (SD): 61.9 ± 12.7 y Gender: 66 M/28 F	X	Cross-Sectional	Clinical interview	**DRF:** In 75.5% Absence of significant differences between M and F RBD <in depressed RBD (*p* = 0.008)
[19]	141 M iRBD Mean age (SD): 66.7 ± 6.7 y 43 F iRBD Mean age (SD): 68.7 ± 7.3 y	X	Cross-Sectional	RBDQ-JP	**Dream-related scores (Factor1):** Absence of significant difference between male and female iRBD
[20]	90 RBD Gender: 63 M/27 F	X	Cross-Sectional	RBDQ-HK	**Dream-related scores (Factor1):** Absence of significant difference between M and F RBD **Vivid dreams:** Absence of significant differences between M and F RBD **Violent dreams:** Absence of significant differences between M and F RBD **Frightening dreams:** Absence of significant differences between M and F RBD
[21]	68 RBD Mean age (SD): 63.7 ± 10.9 y Gender: 49 M/19 F	44 HC Mean age (SD): 62.0 ± 12.2 y Gender: 28 M/16 F	Cross-Sectional	TDQ DTD index	**DRF:** >in RBD compared to HC (*p* < 0.01) >in F RBD compared to HC (*p* = 0.040) Absence of significant difference between M RBD and HC Negative correlated with tonic REM % Absence of significant correlation with phasic REM% **Nightmares recall frequency:** >in RBD compared to HC (*p* < 0.001) >in F RBD compared to HC (*p* = 0.005) >in M RBD compared to HC (*p* = 0.002) Absence of significant correlation with phasic/ tonic REM% **Dream content:** Physically attacked ∘ >in RBD compared to HC (*p* = 0.001) ∘ >in M RBD compared to HC (*p* = 0.006) Snakes, insects: >in RBD compared to HC (*p* < 0.001; uncorrected) Beasts: >in RBD compared to HC (*p* = 0.004) Snakes: >in F RBD compared to HC (*p* = 0.041) Wild, violent beasts: >in RBD compared to HC (*p* = 0.033) Sexual experiences: ∘ >in HC compared to RBD (*p* = 0.030) ∘ >in M HC compared to RBD (*p* = 0.001) Disasters: >in RBD compared to HC (*p* < 0.001; uncorrected) Floods or tidal waves: >in RBD compared to HC (*p* = 0.050) Fire: >in F RBD compared to HC (*p* = 0.044) Paralysis, presences: >in RBD compared to HC (*p* < 0.001; uncorrected) Half-awake/paralyzed: >in F RBD compared to HC (*p* = 0.037) Failure: >in F RBD compared to M RBD (*p* = 0.05) Loss of control: >in F RBD compared to HC (*p* = 0.036) Magic, myth: >in F RBD compared to HC (*p* = 0.010) Seeing yourself as dead: >in M RBD compared to HC (*p* = 0.002) **DTD index:** Absence of significant difference between RBD and HC Absence of significant difference between M RBD and HC >in F RBD compared to HC (*p* = 0.053) <in older RBD and HC compared to younger RBD and HC **Correlations:** Positive correlation between disaster factor and phasic REM% Negative correlation between DTD index and age
[22]	8 non-recallers RBD Age range: 69.8–75.8 y Gender: 6 M/2 F 17 recallers RBD Age range: 66.0–75.0 y Gender: 12 M/5 F	X	Cross-sectional	Dream interview	**DRF:** In 97.3%, RBD in the previous 10 years In 98.6%, RBD during their entire life In childhood: >in recallers RBD compared to non-recallers RBD (*p* = 0.009) In the previous 10 years: >in recallers RBD compared to non-recallers RBD (*p* = 0.0005) In the previous years: >in recallers RBD compared to non-recallers RBD (*p* < 0.0001) Latency to previous dream recall: <in recallers RBD compared to non-recallers RBD (*p* = 0.0005) Frequency per week: >in recallers RBD compared to non-recallers (*p* = 0.0004)
[23]	53 RBD Mean age (SD): 69.0 ± 16.5 y Gender: 39 M/14 F	X	Cross-Sectional Descriptive	Dream questionnaire	**DRF:** >in RBDs in which injury occurred compared to RBDs in which injury did not occur (*p* = 0.002) **Dream content:** Fight theme: absence of significant differences between RBDs in which injury occurred and RBDs in which injury did not occur Chase theme: absence of significant differences between RBDs in which injury occurred and RBDs in which injury did not occur Other themes: absence of significant differences between RBDs in which injury occurred and RBDs in which injury did not occur
[24]	6 RBD Mean age: 54 y Gender: 3 M/3 F	X	Longitudinal (6 weeks) to examine the effect of melatonin treatment	Modification in dream activity after treatment	None of the responders reported any frightening dreams during the treatment period
[25]	8 RBD Mean age: 54 y Gender: 8 M	X	Longitudinal (4 weeks) to examine the effect of melatonin treatment	Modification in dream activity after treatment	**Dream Content:** None of the responders reported having frightening dreams after four days of treatment **DRF:** All patients were able to distinguish placebo from treatment based on a reduction in dream mentation
[26]	39 RBD Mean age (SD): 68.3 ± 7.8 y Gender: 29 M/10 F	X	Longitudinal (28.8 months) to examine the effect of clonazepam treatment	RBDQ-3M	**DRF:** Absence of significant differences pre/post-treatment **Nightmare frequency:** >before than pre-treatment (*p* < 0.01) **Dream-related scores (Factor 1):** >before than pre-treatment (*p* < 0.001) Absence of significant difference between response and no-response group **Dream content changes after treatment:** Decreased violent content after treatment (*p* < 0.01) Decreased frightening content after treatment (*p* < 0.01) Absence of significant changes after treatment in the dreams with emotional or sorrowful content **Correlations:** Positive correlation between dream-related scores (Factor1) and PLMI
[27]	32 RBD Mean age (SD): 61.5 ± 11.1 y Gender: 23 M/9 F	30 HC Mean age (SD): 56.9 ± 16.6 y Gender: 19 M/11 F	Cross-Sectional	Dream questionnaire	**DRF:** Absence of significant between-groups differences **Nightmare distress:** >in iRBD compared to HC (*p* < 0.01) **Dream meaning:** Absence of significant between-groups differences No sex main effects or group × sex interactions for any of the three dream questionnaire subscales **Correlations:** Positive correlation between nightmare distress score and TAS-20 Positive correlation between nightmare distress score and DIF score
[29]	29 iRBD Mean age (SD): 62.9 ± 9.4 y Gender: 23 M/8 F 31 pRBD Mean age (SD): 44.4 ± 9.8 y Gender: 13 M/8 F	31 HC Mean age (SD): 46.0 ± 12.5 y Gender: 11 M/20 F	Cross-Sectional	RBDQ-HK	**Dream-related scores (Factor1):** <in iRBD compared to pRBD (*p* < 0.01) >in pRBD compared to pHC (*p* < 0.01) >in iRBD compared to pHC (*p* < 0.01) **Dream content:** Recurrent nightmares: >in pRBD compared to pHC (*p*< 0.01) Sad theme: <in iRBD compared to pRBD (*p* < 0.05) Angry/agitated theme: >in iRBD and in pRBD compared to pHC (*p* < 0.01) Scary theme: >in pRBD compared to pHC (*p* < 0.05)
[30]	51 iRBD Gender: 41 M/10 F 29 sRBD Gender: 23 M/6 F 27 RBD like Gender: 11 M/16 F	107 HC Mean age (SD): 55.3 ± 9 y Gender: 62 M/45 F	Cross-Sectional	RBDQ-HK	**Dream-related scores (Factor1):** >in RBD compared to HC (*p* < 0.001) >in RBD-like compared to sRBD and sRBD (*p* < 0.005)
[31]	105 RBD Mean age (SD): 67.3 ± 6.4 y Gender: 60 M/49 F	105 HC Mean age (SD): 65.8 ± 5.7 y Gender: 49 M/56 F	Cross-Sectional	RBDQ-KR	**Dream related scores (Factor1):** >in RBD compared to HC (*p* < 0.001)
[32]	13 iRBD Mean age (SD): 66.3 ± 6.5 y Gender: 11 M/2 F	10 HC Mean age (SD): 62.3 ± 7.5 y Gender: 7 M/3 F	Cross-Sectional	RBDQ-KR	**Dream-related scores (Factor1):** >in RBD compared to HC (*p* < 0.001) Absence of significant correlation between Factor 1 and power spectral density changes during phasic and tonic REM sleep in RBD
[33]	94 RBD Mean age (SD): 67.6 ± 7.3 y Gender: 53 M/41 F	50 HC Mean age (SD): 65.4 ± 6.0 y Gender: 24 M/26 F	Cross-Sectional	RBDQ-KR	**Dream related scores (Factor1):** >in RBD than HC (*p* < 0.001) **Correlations:** Negative correlation between Item 2 (RBDQ-KR) and the CERQ adaptive score Absence of significant correlations between CERQ adaptive score and emotional, violent, aggressive, or frightening dreams
[48]	12 RBD Mean age (SD): 65.6 ± 10.7 y Gender: 11 M/1 F	12 HC Mean age (SD): 63.3 ± 12.9 y Gender: 8 M/4 F	Cross-Sectional	Dream diary (3 weeks)	**Dream content:** Absence of significant difference between RBD and HC Absence of significant difference between RBD with normal dreams and with dreams associated with motor behavior **Threat simulation dream content:** Absence of significant difference between RBD and HC Absence of significant difference between RBD with normal dreams and with dreams associated with motor behavior **Bizarreness Density Index:** Absence of significant difference between RBD and HC **Words number:** Absence of significant difference between RBD and HC
[49]	13 clonazepam-treated iRBD Mean age (SD): 65.3 ± 10.9 y Gender: 12 M/1 F 11 untreated iRBD Mean age (SD): 68.9 ± 6.8 y Gender: 9 M/2 F	12 HC Mean age (SD): 63.3 ± 12.8 y Gender: 8 M/4 F	Cross-Sectional	Dream diary (3 weeks) scored by HVdC TSS	**Dream Reports (*n* 214):** In total, 92 in the clonazepam-treated iRBD group In total, 70 in the untreated iRBD group In total, 52 in the HC group **Dream reports associated with DEBs** In total, 43% (*n* 40) in the clonazepam-treated iRBD group In total, 64% (*n* 45) in the untreated iRBD group In total, 0% (*n* 0) in the HC group **Dream reports without DEBs:** In total, 56% (*n* 52) in the clonazepam-treated iRBD group In total, 36% (*n* 25) in the untreated iRBD group In total, 100% (*n* 52) in the HC group **Dream content:** Frequency of threatening dream contents: absence of significant between-groups differences Threatening events ∘ In the clonazepam-treated iRBD group: absence of significant differences ∘ In the untreated iRBD group: absence of significant Frequency of Friendliness item: >in treated and untreated iRBD groups compared to HC group (*p* = 0.036) Frequency of Aggressive dream contents ∘ Absence of significant between-groups differences ∘ In the clonazepam-treated iRBD group: >in dream reports associated with DEBs than without DEBs (*p* = 0.007) ∘ In the untreated iRBD group: >in dream reports associated with DEBs than dream reports without DEBs (*p* = 0.012) Frequency of Familiar Figures ∘ In clonazepam-treated iRBD group: >in dream reports associated with DEBs than without DEBs (*p* = 0.014)
[34]	123 RBD divided in 96 with treatment- improvement Mean age (SD): 65.7 ± 8.5 y Gender: 61 M/35 F 27 without treatment- improvement Mean age (SD): 66.1 ± 7.5 y Gender: 15 M/12 F	X	Longitudinal (17.7 months) to examine the effect of clonazepam treatment	RBDQ-KR	**Dream-related scores (Factor1):** Absence of significant difference between responding and no-respond groups

Abbreviations: Aggression/Characters ratio (A/C); Aggression Questionnaire (AQ); Continuous Positive Airway Pressure (CPAP); Dream Enactment Behaviors (DEB); Dream Recall Frequency (DRF); Dream Theme Diversity (DTD); Female (F); Hall and Van De Castle method (HVdC); Healthy Controls (HC); idiopathic REM sleep Behavior Disorder (iRBD); Male (M); Modified RBD Questionnaire (RBDQ-3M); Periodic Limb Movement Index (PLMI); Psychiatric REM sleep Behavior Disorder (pRBD); Rapid Eye Movements (REM); REM sleep Behavior Disorder (RBD); RBD Questionnaire–Japanese version (RBDQ-JP); RBD Questionnaire—Hong Kong version (RBDQ-HK); RBD Questionnaire—Korean version (RBDQ—KR); secondary REM sleep Behavior Disorder (sRBD); Threat Simulation Scale (TSS); Typical Dreams Questionnaire (TDQ); Years (y).

Another key feature of this REM parasomnia is the prevalence of the disorder in the male population [10,54], reporting more severe symptoms and nocturnal behavioral episodes in men than women with RBD [41,55]. However, studies that explored gender differences in oneiric activity revealed the absence of significant differences between males and females in dreams and nightmares recall rates [14,18,19], vividness [20], and contents [19,20].

Moreover, elevated dream and nightmare recall frequency (from 98.6% to 75%) was reported in RBDs [12,14,18,21,22]. In 63% of dream reports, the recall was associated with behavioral episodes [13], and higher DRF was found in RBD patients causing injuries than in RBDs in which injury did not occur, although no between-groups differences were reported in the dream contents [23]. However, this peculiar framework of oneiric activity observed in RBD, characterized by high DRF and violent dream contents, has not always been confirmed by studies that examined the effect of treatment on RBD symptoms and studies that compared patients with HCs.

On the one hand, we can affirm that specific dream contents, characterized by violent and aggressive themes, are typical of RBD. Moreover, longitudinal studies showed that melatonin and clonazepam assumptions suspended frightening, violent dreams [24,25,26], and nightmares [26] during treatment. In the same line, compared to HCs, dreams in RBDs were characterized by a prevalence of violent and aggressive themes, also involving animal or people attacks [16,21,27,28,29], with a high incidence of negative emotions and nightmare distress [16,27]. Moreover, using the RBD Questionnaire (RBDQ) [30], patients showed higher scores in Factor 1 (which considers the dreams and nightmares frequency and the emotional, violent, and aggressive contents) than HCs [29,30,31,32,33]. This evidence has been explained in two ways. The first hypothesis regards the biological and evolutionistic role of dreaming in simulating dangers and threats that have to do with ancestral human fears to “prepare” the subject to rehearse threat perception and its avoidance during wakefulness [42]. The second hypothesis to explain aggressive features in the RBDs’ dream reports could be to account for cognitive dysfunctions due to impairments observed in the frontal cortex [56]. Thus, these results suggest that violent nightmares and dreams in RBD may have clinical importance in predicting the possible onset of neurodegeneration. Indeed, the aggressive dream contents reported by PD patients is suggestive to be related to frontal cognitive dysfunction [57]. In addition, the violent and aggressive dream contents experienced by RBDs could be also explained by the lack of inhibition due to frontal cortex dysfunctions, leading to archaic defense behaviors acted out in dreams [56]. However, findings that may help to clarify this relationship will be reported and discussed in the next paragraph.

On the other hand, studies reported the absence of more vivid dreams in RBDs [28], no higher DRF [27,28,47], and no differences in dream contents [28,48] when comparing RBDs to HCs and when comparing pre and post treatment [34,49] (See Table 1 and Table 3). Contrasting results between studies in RBDs can be explained by methodological limits that affect results. Indeed, retrospective studies assessing oneiric activity in the past and during the entire patient’s life reported high rates of dreams and nightmare recall [12,13,14,18,21,23]. Retrospective methodology to collect dreams leads to the so-called “recall bias”, which is the predisposition of patients suffering from RBD to recall more frequently vivid dreams with violent and frightening contents accompanied by motor behaviors [58].

Concluding, although literature findings confirm a predisposition of RBDs to report oneiric activity characterized by violent content, findings in dream frequency are not sufficient and are not solid enough to conclude a clear increase of DRF in RBDs. We recommend employing prospective experimental designs to collect dreams in future studies exploring dreaming in RBDs.

### 3.2. Dreaming in RBD: A Window into Neurodegenerative Mechanisms?

The intrinsic features of dream activity in iRBD patients described in the previous paragraph focus researchers’ attention on identifying potential markers able to predict the phenoconversion of parasomnia into α-synucleinopathies years before signs of neurodegeneration emerge [6]. Indeed, iRBD and PD conditions share similar dream features, such as a high prevalence of dreams with violent and distressing contents, and nightmares [59]. In this view, establishing a relationship between oneiric activity in iRBD and neuropathological mechanisms could allow early detection of neurodegeneration processes and make it possible to understand neural mechanisms underlying the generation and recall of dreams.

Our literature analysis reported 6 studies investigating dreaming in RBDs compared to PDs [35,36,46,50,51,52] and one study that compared dreaming between RBDs and DLBs [37] (See Table 2).

**Table 2 jcm-11-06379-t002:** Sample, design, tools, and findings in studies investigating dreaming in RBD with and without neurodegenerative disorders.

Study	Sample	Design	Dream Measures	Main Findings
[46]	49 PD + RBD Mean age (SD): 68.3 ± 7.5 y Gender: 33 M/16 F 36 PD—RBD Mean age (SD): 69.9 ± 9.6 y Gender: 18 M/18 F 30 HC Mean age (SD): 66.8 ± 9.9 y Gender: 12 M/18 F	Cross-Sectional	Dream Diary (1 month) scored by HVdC	**Total dreams collected: 106** **DRF** <2 dreams per month Absence of significant difference **Dream content** Dreamer as a character: absence of significant difference between PD + and—RBD Sex of dreamed subjects: absence of significant difference between PD + and—RBD Identity of dream characters ∘ Familial persons: <in PD + RBD than PD—RBD (*p* < 0.001) ∘ Known persons: absence of significant difference ∘ Unknown people: absence of significant difference ∘ Animals: absence of significant difference ∘ Undefined: >in PD + RBD than PD—RBD (*p* < 0.001) Success: absence in the dream reports Failure: absence of significant difference Good fortune: absence of significant difference Misfortune: absence of significant difference **Emotional content** Apprehension: <in PD + RBD than PD—RBD (*p* = 0.029) Sadness: absence of significant difference Happiness: absence of significant difference Confusion: absence of significant e difference Anger: absence of significant difference **Aggressive dream content** Mean level of reported aggressiveness: a trend of >in PD + RBD than PD—RBD Aggressive act which results in the death of a character: absence of significant difference Aggressive act which involves an attempt to physically harm a character: absence of significant difference Aggressive act which involves a character being chased, captured, confined, or physically coerced into performing some act: absence of significant difference Aggressive act which involves the theft or destruction of possessions belonging to a character: absence of significant difference Aggressive act in which a serious accusation or verbal threat of harm is made against a character: absence of significant difference Situations where there is an attempt by one character to reject, exploit, control, or verbally coerce another character: absence of significant difference Aggression displayed through verbal or expressive activity: absence of significant difference Covert feeling of hostility or anger without any overt expression of aggression: absence of significant difference
[50]	65 PD + RBD Mean age (SD): 65 ± 9 y Gender: 44 M/21 F 35 PD—RBD Mean age (SD): 61 ± 13 y Gender: 21 M/14 F	Cross-Sectional	Interview	**Dream Content in PD + RBD** Fighting or running/fleeing: 54% **Dream reports examples** “I am a police duck, flying after a pigeon-thief” “I am dreaming that I am singing as I used before PD” “I am on canoe, attacked by caimans, trying make them flee” “I am flying lying on my back with the feet in front and I am breaking with my feet” “I am rehearsing a speech for the town council” “I am a knight fighting with a foil to saving my endangered lady-love”
[51]	6 PD + RBD Mean age (SD): 58.5 ± 8.4 y	Multiple Awakenings	Dream Questionnaire	**DRF** In 25 of the total 35 awakenings (71.4%) Mean DRF: 71.4 ± 31.8% (range 14–100%) In 17 awakenings (48.6%) both REM-related movements and dream recall were present simultaneously Minor movements and twitching occurred in conjunction with dream recall in 4 REM episodes Moderate movements were manifest with dream recall in 9 REM episodes (17% of all awakenings) Violent movements coinciding with dream recall were present in 4 (11%) of the REM awakenings In the remaining awakenings with dream recall (*n* = 8), no movements were observed during preceding REM sleep No significant difference in the presence of movements when stratified for whether or not dream recall was present **Judge performance:** despite the presence of positive emotions in 4 dream reports, the 4 most accurately matched dream-video pairs were the ones with negative dream emotions
[52]	9 PD + RBD 1 PD 3 HC	Cross-sectional Descriptive	Immediate Free Dream Recall	When awakened during REM behavioral episodes, all 9 RBD patients reported vivid but non-threatening dreams 7 RBD patients accurately describe their dreams
[35]	36 PD + RBD Mean age (SD): 67.2 ± 7.3 y Gender: 25 M/11 F 26 PD—RBD Mean age (SD): 68.3 ± 10 y Gender: 18 M/8 F 24 PD + probable RBD	Cross-Sectional	NMSQuest—Item 24	**Intense, vivid dreams:** >in PD + RBD compared to PD—RBD (*p* < 0.001) **Distressing dreams:** >in PD + probable RBD compared to PD—RBD
[36]	9 PD + RBD Mean age (SD): 61.2 ± 9.8 y Gender: 7 M/2 F 8 PD—RBD Mean age (SD): 64.0 ± 10.3 y Gender: 6 M/2 F	Multiple Awakenings	Semi-Structured Interview Immediate Dream Recall	**DRF changes after the PD onset** Changes did not differ between groups <DRF in 46.7% >DRF in 26.7% No changes in 20.0% **DRF in the sleep lab** Dream recall was better from REM than NREM sleep No significant between-group differences in dream recall from REM or NREM sleep Equally frequent after spontaneous (50%) and forced awakenings (55%) in both groups **Nightmare recall frequency:** >in PD + RBD (*p* = 0.008) **Awakenings** PD + RBD: on 36 awakenings in conjunction with behaviors recalled dreams in 23 of these awakenings Total dreams acquired: 69 37 in PD + RBD participants 32 in PD—RBD participants No significant differences in the length of dream reports **Dream content** Dream content changes after the PD onset >vivid and negatively toned dreams in 5 PD + RBD (55.6%) ∘ >negative dreaming in 1 PD—RBD ∘ The rest had not observed any changes ∘ Absence of significant between-groups differences ∘ Nature and intensity of action-filledness: absence of significant between-groups differences ∘ Outwardly expressed action elements were >prevalent than environmental events in both groups ∘ Intensity of the elements describing action-filledness: >often evaluated as low (59.2%) than moderate (25.3%), and least often as intense (15.4%) Vividness ∘ Absence of significant between-groups differences ∘ Intensity of the elements describing vividness: >often evaluated as low (59.2%), then as moderate (25.3%), and least often as intense (15.4%) Threatening events and their quality ∘ On average: 4.6 threatening events per dream ∘ Number of threatening events: absence of significant difference Type of threat ∘ Failure to achieve a set goal (37.5%) ∘ Aggression (25.0%) ∘ Accident and Illness (15.6%) ∘ Catastrophe (3.1%) Target of threat ∘ The dreamer himself (71.9%) ∘ A significant other or resources (18.6%) ∘ A non-significant other (15.6%) ∘ Non-significant resources in 9.4% Severity of threat ∘ Minor (50.0%) ∘ Life-threatening (15.5%) ∘ Threatened the physical well-being of the dream self (6.3%) Reaction to the threat ∘ Reasonably and appropriately (65.6%) ∘ Not scored due to interruption of the dream situation (34%) Nature of the threat ∘ Realistic (78.1%) ∘ Realistic but improbable (21.9%) **Emotions** ∘ Negative emotional tone: >often reported than positive or balanced dreams, or dreams lacking emotional valence ∘ No between-group differences in the distribution of emotional valence of dream reports ∘ In PD + RBD > negative than positive dreams
[37]	13 DLB Mean age (SD): 78.4 ± 6.2 y Gender: 6 M/7 F 13 DLB + RBD Mean age (SD): 77.4 ± 5.7 y Gender: 10 M/3 F	Cross-Sectional Descriptive	Clinical Interview	**Unpleasant dream recall:** 7 of the 13 (53.8%) patients with DLB + RBD

Abbreviations: Dementia with Lewy Bodies (DLB); Female (F); Hall and Van de Castle method (HVdC); Healthy Controls (HC); Male (M); Non-Motor Symptoms Questionnaire (NMSQuest); Parkinson’s Disease (PD); Rapid Eye Movements (REM); REM sleep Behavior Disorder (RBD); Years (y).

These studies described high rates (from 50% to 70%) of DRF [37,51,52] and more vivid dreams [35,36,52] in RBD conditions in comorbidity with α-synucleinopathies (i.e., PD and DLB) than RBD without signs of neurodegeneration.

To clarify the association between dream-related features and neurodegenerative mechanisms that seem to underlie the RBD pathophysiology, interesting findings are those that consider specific RBD symptoms as RSWA. Results showed that in RBD the higher percentage of phasic muscle activity during REM sleep was related to more elements with natural disasters in dream recall. On the other hand, RBDs that showed higher tonic muscle activity percentage during REM sleep were less prone to recall dreams [21]. RSWA is one of the key criteria to diagnose RBD syndrome [1] and authors suggest it may reflect a progressive damage to the brainstem [43]. Tonic and phasic EMG activities during REM sleep have different neural mechanisms. Specifically, phasic EMG activity is regulated by locomotor nuclei in the ventromedial medulla, structure impaired yet in the early stages of PD [60]. On the other hand, increasing tonic muscle activity, depending on REM-on neurons of the sublaterodorsal tegmental nucleus, seems to be strictly associated with the phenoconversion to PD [61]. However, evidence from this review is not enough to confirm a strong relationship between dreams and RSWA features.

Moreover, a single study [36] retrospectively explored changes in RBDs converted in PD, reporting higher nightmares frequency after the onset of PD symptoms. However, a paper [46] investigating DRF through sleep diaries compiled for one month showed lower rates of dreams reported in PDs with and without RBD (<2 dreams per month) and no significant differences in DRF between the two groups. In the same direction, no between-group differences are revealed in the dream contents reported by iRBDs and RBDs with PD symptoms. Specifically, the absence of significant differences was found in vividness and intense emotional contents such as threat, aggression, or negativity [36,46]. Also in this case, these findings may be explained by the “recall bias” occurring when dreams are collected retrospectively. Indeed, studies reporting the absence of group differences in DRF and contents between RBD patients with and without PD adopted prospective designs (i.e., daily dream logs for one month [46] and systematic laboratory awakenings protocol [36]). Undoubtedly, these procedures reduce recall bias. These findings are consistent with the work by D’Agostino et al. [48] illustrated in the previous paragraph, which compared dream contents in RBDs and HCs employing immediate free recall through 3 weeks of daily dream diaries.

Overall, although the theory proposed is fascinating, the state of the art does not confirm the predictive value of dream features as markers able to track the neurodegenerative process in RBD.

### 3.3. Dream Features in RBD and NREM Parasomnias or Other Sleep Disorders

RBD nocturnal episodes appear as abnormal motor and vocal behaviors (i.e., punching, falling out of bed, and shouting) often associated with peculiar dream mentation [4]. These behavioral manifestations may also occur in other sleep disorders (e.g., OSA) and other NREM parasomnias (e.g., SW and ST). In severe OSA conditions, quite common among older adults, the respiratory effort and/or breathing resumption associated with sleep arousals lead to motor and vocal behaviors both during REM and NREM sleep [38]. In clinical settings, it is common to perform a differential diagnosis between RBD and OSA, based on the RSWA as a key feature for the RBD diagnosis [1].

Regarding SW and ST, these NREM parasomnias show complex, unaware, and aggressive or harmful motor and vocal behaviors, which might be mistaken for RBD episodes. However, SW/ST behaviors occur mostly during the first half of the night and always during Slow Wave Sleep (SWS) [1]. Consequently, the vPSG is the gold standard for a differential diagnosis between RBD and other sleep disorders.

Moreover, since the importance of operating a rapid and effective differential diagnosis to address different treatments and to indicate different prognoses, clinical information may also be useful. In this vein, the associated dream mentation features to the behavioral episodes may be suitable in terms of clinical implication to differentiate RBD from sleep disorders that mimic RBD symptoms.

In the literature, two studies [39,62] compared dreaming between RBD and OSA patients, and three studies [17,28,47] explored dreaming in RBDs and SWs or STs. As shown in Table 3, findings suggest little relevance of DRF as a signature of RBD.

**Table 3 jcm-11-06379-t003:** Sample, design, tools, and findings in studies investigating dreaming in RBD and other sleep disorders.

Study	Sample	Design	Dream Measures	Main Findings
[17]	24 RBD Mean age (SD): 68.6 ± 8.8 y Gender: 19 M/5 F 32 SW/ST Mean age (SD): 31.4 ± 8.4 y Gender: 16 M/16 F	Cross-sectional	Immediate free dream recall scored by HVdC TSS Dream complexity	**No. of dreams:** During the lifetime: <in RBD compared to SW/ST (*p* = 0.04) In the sleep lab: absence of significant differences **N° of words in the dream report:** During the lifetime: >in RBD compared to SW/ST (*p* = 0.07) In the sleep lab: <in RBD compared to SW/ST (*p* = 0.03) **Immediate DRF in the sleep lab:** In total, 25% in RBD **Complexity:** During the lifetime: >in RBD compared to SW/ST (*p* = 0.006) In the sleep lab: <in RBD compared to SW/ST (*p* = 0.05) **Bizarreness:** During the lifetime: <in RBD in the total score (*p* = 0.03) and in the type 4 (*p* = 0.04) compared to SW/ST In the sleep lab: absence of significant differences **Dream content during the lifetime:** Aggression and violence: >in RBD than SW/ST (*p* = 0.04) Accidents and misfortunes: <in RBD than SW/ST (*p* = 0.008) Target of the threat: absence of significant differences Participation in the dream itself categories: absence of significant between-group differences **Dream content in the sleep lab:** Target of the threat: individuals important to subject <in RBD compared to SW/ST (*p* = 0.06) No significant differences in all other categories
[28]	64 RBD Mean age (SD): 68.6 ± 8.0 y Gender: 44 M/20 F 62 SW Mean age (SD): 31.7 ± 9.5 y Gender: 29 M/33 F 66 oHC Mean age (SD): 67 ± 7.8 y Gender: 43 M/23 F 59 yHC Mean age (SD): 31.9 ± 9.3 y Gender: 29 M/30 F	Cross-Sectional	RBDSQ	**DRF:** absence of between-group differences **Vivid dreams:** absence of between-group differences
[47]	20 RBD Mean age (SD): 66.5 ± 6.5 y Gender: 16 M/4 F 19 SW Mean age (SD): 34.4 ± 15.4 y Gender: 6 M/13 F 18 HC Mean age (SD): 57.9 ± 5.3 y Gender: 14 M/4 F	Cross-Sectional	Immediate free recall	**DRF:** absence of significant between-group differences
[39]	16 iRBD Mean age (SD): 64.5 ± 5.1 y Gender: 13 M/3 F 16 OSA Mean age (SD): 59.6 ± 7.7 y Gender: 11 M/5 F 20 HC Mean age (SD): 63.0 ± 9.8 y Gender: 16 M/4 F	Cross-Sectional	Not specified	**Unpleasant dream content:** Attacked by someone ∘ OSA: 62.5% ∘ RBD 93.8% Chased by someone ∘ OSA 62.5% ∘ RBD 81.3% Arguing with someone ∘ OSA 50% ∘ RBD 68.8% Falling abruptly ∘ OSA 25% ∘ RBD 68.8% Attacked by animals ∘ OSA 25% ∘ RBD 43.8%
[62]	118 RBD Mean age (SD): 66.5 ± 8.4 y Gender: 91 M/27 F 106 OSA Mean age (SD): 61.6 ± 8.4 y Gender: 57 M/49 F	Cross-Sectional	RBDQ—Beijing	**Dream related scores (Factor 1)**: >in RBD compared to OSA (*p* < 0.001)
[6]	15 RBD + PTSD Mean age: 55.2 y 12 RBD Mean age: 57.6 y 7 PTSD Mean age: 56.7 y	Cross-Sectional	Not specified	**Dream content/emotions:** Fright, *n* (%) ∘ RBD + PTSD: 15 (100%) ∘ RBD: 8 (67%) ∘ PTSD: 7 (100%) Pleasure *n* (%) ∘ RBD + PTSD: 0 ∘ RBD: 2 (17%) ∘ PTSD: 0 Unsure *n* (%) ∘ RBD + PTSD: 0 ∘ RBD: 2 (17%) ∘ PTSD: 0 **Dreams related to past trauma:** ∘ RBD + PTSD: 15 (100%) ∘ RBD: 5 (42%) ∘ PTSD: 7 (100%)

Abbreviations: Dream Recall Frequency (DRF); Female (F); Hall and Van De Castle method (HVdC); Healthy Controls (HC); idiopathic REM sleep Behavior Disorder (iRBD); Male (M); old Healthy Controls (oHC); Post-Traumatic Stress Disorder (PTSD); REM sleep Behavior Disorder (RBD); REM sleep Behavior Disorder Screening Questionnaire (RBDSQ); Sleep Terrors (ST); Sleep-Walkers (SW); Threat Simulation Scale (TSS); Years (y); young Healthy Controls (yHC).

However, these works indicate in RBDs a prevalence of unpleasant and complex dreams [17,28,39,62], specifically containing attacks and violent contents [17,39]. However, this evidence refers to all retrospective dream collection. Indeed, when dream recall was performed immediately after the awakening in the sleep laboratory, these findings were completely reversed, showing in SWs/STs more complex and long dream reports, without significant difference in violent and unpleasant dreams [17].

One particular case regards sleep disorders due to trauma or severe anxiety states, such as PTSD. Despite PTSD not being considered a sleep disorder by the ICSD-3 [1], sleep-related symptoms are common. Specifically, PTSD patients report sleep disturbances, hyperarousal, and sleep movements. Furthermore, intrusive thoughts and images are key features of the PTSD diagnosis, which occur as nightmares during the night [63]. As in RBD, an increased phasic and tonic electromyography (EMG) activity during REM sleep can be observed also in PTSD [64,65,66], caused by similar neuroanatomic abnormalities in both syndromes. Indeed, one of the hypotheses advanced points to a loss of neurons in the locus coeruleus in patients with RBD and with PTSD [40]. Despite RBD and PTSD sharing such clinical similarities, only one study in literature [67] described dream content between these two conditions. Although the authors provided only a descriptive overview not performing any statistical analysis, nevertheless the findings reported seeming relevant. Indeed, 100% of PTSDs with and without RBD recalled dreams containing frightening emotions and unpleasant dreams related to past trauma; on the other hand, RBD patients without PTSD symptoms reported lower rates of frightening dreams (67%) and dreams related to trauma (42%). Conversely, pleasant dreams were reported in 17% of RBDs and never reported by PTSDs (with and without RBD).

This preliminary evidence suggests that nightmares are a PTSD hallmark, beyond the presence of RBD symptoms. However, further works investigating the relationship between nightmare occurrence and EMG activity in these two disorders would be interesting in order to consider similar neuropathological mechanisms underlying RBD and PTSD.

### 3.4. RBD in Infants

For a long time, it was thought that RBD was a parasomnia affecting particularly elderly men. Nevertheless, a similar prevalence in women [68] and in all ages has been observed over time. In this regard, RBD was also found during childhood and adolescence.

Although most of the clinical features of RBD in older adults also occurred in children, in this last population, specific characteristics are found. Case reports showed that most children suffering from RBD also showed other neurological (i.e., cerebellar tumor [69,70], juvenile PD [71], narcolepsy [9,72,73,74]) and neuropsychological (i.e., autism [9,75], anxiety, depression, obsessive-compulsive disorder, attention deficit hyperactivity disorder [9]) disorders. However, many of the main clinical aspects of RBD in pediatrics are not fully known. Indeed, the outcome and the course of this parasomnia in children are unclear since follow-up studies aimed to trace its clinical evolution are still lacking.

Furthermore, assessing symptoms and general clinical features may be quite challenging in this population because of communication problems due to the early age or concomitant handicap conditions [76]. This limitation is particularly relevant in collecting subjective sleep symptoms or self-reported dream contents. In fact, a unique study in the literature [9] assessed dream activity in RBD children, declaring the failure in collecting dream contents in two subjects because they were unable to describe it (See Table 4). However, findings in children confirmed evidence reported in older RBDs, showing high rates of nightmares and vivid frightening dreams involving violence or chasing. Moreover, also in children, the clonazepam treatment leads to the resolution of RBD symptoms, including nightmares.

**Table 4 jcm-11-06379-t004:** Sample, design, tools, and findings in the study investigating dreaming in children with RBD.

Study	Sample	Design	Dream Measures	Main Findings
[9]	15 RBD Mean age: 9.5 y Gender: 11 M/15 F	Retrospective Descriptive Case Series	Not specified	**Dream Content** Nightmares (*n* 13) Vivid frightening dreams involving violence or chasing (*n* 10) *n* 2 had speech apraxia and unable to describe dream contents Resolution of nightmares after clonazepam treatment (*n* 10)

Abbreviations: Female (F); Male (M); REM sleep Behavior Disorder (RBD); Years (y).

Notably, we owe the current knowledge about oneiric activity in children with RBD to a single study [9]. Although infant RBD is a rare condition, further studies, especially those involving longitudinal design, will help understand the pathophysiology behind this condition and the long-term implications of childhood RBD.

## 4. Conclusions and Future Perspectives

To sum up, the results reported in our review suggest a double interpretation of dreaming in RBD, depending on the design adopted by the studies: retrospective or prospective. Indeed, retrospective studies mainly point to RBD as characterized by unpleasant dreams and nightmares, containing animal or people attacks, violence, and negative emotions. These features arise mostly when RBD patients are compared to patients with other parasomnias, such as SW and ST, and patients with neurodegenerative symptoms. This evidence suggests a potential clinical relevance of aggressive contents and high DRF in the pathophysiology of RBD and a potential role of oneiric activity as a marker to track neurodegenerative processes.

However, prospective studies do not confirm this framework, suggesting a similar oneiric activity in idiopathic and secondary RBD, and between RBD and other parasomnias. The discussion of findings leads us to the “recall bias” phenomena, which could obscure the potential association between oneiric features and this REM parasomnia. Moreover, despite that there has been a surge in research about several aspects of RBD in recent years, from this review it can be noted that there are few studies in the literature aimed at exploring dream activity in RBD. Although oneiric activity is a central feature of RBD, most of the studies discussed in this paper assessed dreaming in patients without standardized protocols and only with a descriptive approach.

These methodological limitations bring out the need to deepen the issue of dreaming in RBD. Thus, we believe that further steps in this research area should be done in future studies considering (a) the application of robust experimental protocols and prospective tools to collect dreams; (b) the relationship between dream features in RBD and the motor manifestations or the EMG activity features during REM sleep; (c) how treatment for RBD symptoms affects oneiric activity; (d) dream features in other populations in which RBD is less frequent, but still presents, such as in female patients and children; (e) oneiric features as potential indexes to operate differential diagnosis between RBD and other disorder that mimic this REM parasomnia.

Overall, investigations in these directions, applying more controlled experimental designs, will offer relevant clinical insights. Indeed, in a translational view, dream research, which until now has been a niche of empirical research, could provide knowledge about RBD useful in clinical settings. Indeed, although available data are still not robust enough, in the future, dream features in RBD could help clinicians to monitor the severity of the disease and the possible conversion in synucleinopathies, but also to operate a differential diagnosis between RBD and other parasomnias. Our work suggests the relevance of considering dream features in clinical settings, supervised by general practitioners and sleep specialists. Indeed, monitoring the dream frequency and the dream contents with negative valence may be useful to track the presence of comorbidities between RBD and nightmare disorders for a first and low-cost screening. Moreover, based on the continuity hypothesis, which suggests a permanence between wake and sleep thoughts [77], considering the relationship between violent and frightening dreams and waking experiences could improve the patient’s well-being. Moreover, since changes in dream recall seem dependent on cognitive deterioration [78], monitoring alterations in the DRF in RBD patients may be useful to evaluate the relationship between the frequency of dream recalls and the neurodegenerative processes onset.

Despite the promising translational value of dreams features in clinical settings, these tips should be considered with caution, given these data’s novelty and weak points.

## Figures and Tables

**Figure 1 jcm-11-06379-f001:**
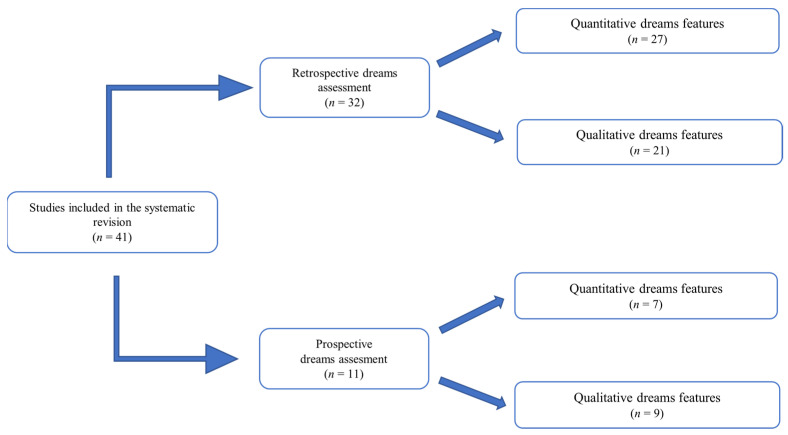
Studies categorization. Among 41 studies analyzed, 32 studies performed retrospective dreams assessment [9,11,12,13,14,15,16,17,18,19,20,21,22,23,24,25,26,27,28,29,30,31,32,33,34,35,36,37,38,39,40] extracting quantitative [4,12,13,14,16,17,18,19,20,21,22,23,25,26,27,28,29,30,31,32,33,36,37,38,39,40] and qualitative [9,11,13,15,16,17,19,20,27,28,30,31,32,33,34,35,36,40,41,42,43] dream features. 11 studies performed prospective dreams assessment [17,36,44,45,46,47,48,49,50,51,52] extracting quantitative [17,36,46,47,48,49,51] and qualitative [17,36,44,45,46,48,49,50,52] dream features.

## Data Availability

Not applicable.

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
