# Peer review of "Dreaming in Parasomnias: REM Sleep Behavior Disorder as a Model"

_jcm, 2022, doi:10.3390/jcm11216379_

Round 1
Reviewer 1 Report
Thank you for opportunity to review the paper „ Dreaming in parasomnias: REM sleep Behavior Disorder as a Model .The manuscript is very well written, however there are few shortcomings and minor issues.
1) Have authors followed the PRISMA (Preferred Reporting Items for Systematic Reviews and Meta-Analyses)?
2) Did authors exclude cases and case series?
3) Was the paper reported in any international register of reviews?
4) How many authors performed literature review? What was the access date?
5) The clues for GPs or sleep specialists should finally be presented .
Reviewer 2 Report
This is a well-written review by an experienced team of authors. I have no concerns with the methodology, writing, structure, references, or ethics of the work. Unlike many such reviews, I found this paper to have good narrative structure, and that it was a pleasure to read. Those in the field will find this a valuable contribution to the literature.
There are several exceptionally minor grammatical errors that can be addressed through editorial discretion, which I assume will be noted on further proofing.
Reviewer 3 Report
Good topic review - few suggestions:
Line 79 - quantitative examination ..... - change to quantitative/qualitative evaluation
Line 133 - higher DRF was found in - change to higher DRF was found in RBD
Line 141 - assumptions .... change to addition
Line 148 - The first one... change to the first hypothesis
Line 155- reported by PD patients seem ... change to reported by PD patients is suggestive
Line 205 - and reflects ... change to author suggests it may reflect
Line 209 to 211 - reference 10 does not seem to suggest what you have quoted - please review and let/s know the source of that assertion
Line 271-273 - where is the reference that documents that both RBD & PTSD have definitive loss of neurons in the locus ceruleus - it is a hypothesis
Reviewer 4 Report
Comments to the Author:
This manuscript presents important and up to date review on dreaming in parasomnias. I have just minor issues that need to be addressed:
1. Line 49 – A brief description of isolated RBD is advisable
2. Line 93 – what you mean by 174 investigations, number of subjects with RBD?
3. Lines 103-104 – The sum of “retrospective (n = 32) and prospective (n = 11)” studies is more than 41, a number of included studies (line 87)
4. Figure 1 – Again, number of studies does not match 41, 32, 11
5. Line 164 – “affect studies”, better “affect results”
6. Line 172 – Authors report that “literature findings confirm a predisposition of RBDs….”, and then conclude that it’s not enough “to conclude a clear increase of DRF in RBDs. There is some contradiction between these statements.
7. Table 1 – It is difficult to track which studies have a control group. If possible, include separate column to indicate that information.
8. Table 1 – study 18 – has 56 RBD patients, and authors present dream reports examples from 9 subjects. For a table 9 examples are too much; moreover, it is not clear how these 9 subjects were selected, is there any specific reasons to present these 9 examples and not 3 or 10?
9. Table 1 – study 35 is missing
10. Table 2 – For study 51 authors describe the sample as RBD+PD, and for the remaining studies as PD+RBD; Is the sequence of RBD and PD considered in the table?
11. Lines 247-248 – singular / plural is mixed for description of behaviors.
